# Semi-supervised Diffusion Solver for Travelling Salesman Problem

## Abstract

We propose a semi-supervised diffusion solver for solving the Travelling Salesman Problem (TSP). Data-driven combinatorial optimization models recently attract an amount of attention, since they have shown promising results in solving various NP-hard problems without too much expert knowledge. However, most of them rely on reinforcement learning (RL) and supervised learning (SL) which face some intractable challenges: RL methods often encounter sparse reward problems and SL methods pose a strict assumption that the optimal solution (label) is always available. To address these challenges in arbitrarily large-scale TSP, this article proposes a novel semi-supervised learning-based diffusion framework towards a more general situation, *i.e.*, we can freely produce instances as many as possible but the acquisition of optimal solution is costly. This semi-supervised paradigm is made viable by modeling the generative process upon a special transition matrix, which facilitates the effective learning of the generative diffusion, compared with learning the heatmap directly like other solvers do. Comprehensive experiments validate our method across various scales TSP, showing that our method remarkably outperforms state-of-the-art data-driven solvers on large benchmark datasets for Traveling Salesman Problems, and has an outstanding generalization ability.

## 1 Introduction

Combinatorial Optimization (CO) problem is a classical mathematical problem which aims to find the optimal solution in a discrete space. It is still a challenge in computer science, since a large portion of CO problems are NP-hard and cannot be solved in polynomial time with traditional solvers which are very complicated and require significant expert knowledge Arora (1996). The traveling salesman problem (TSP) is a well-known CO problem which has numerous real-life applications, including transportation, robots routing, biology, circuit design and so on Vesselinova et al. (2020). For TSP, given $N$ cities and the Euclidean distances between each pair of cities, we try to find the cheapest or shortest tour whose cost is minimized over the input graph in which the tour starts from a beginning city and visits each node exactly once before finally returning to the beginning city. During the past decades, most researchers pay a large attention to designing more efficient heuristic solvers Arora (1996); Gonzalez (2007) to approximate optimal solutions.

Recently, with the development of deep learning, more and more data-driven approaches have been proposed and shown promise in solving CO problems without too much manual intervention or specific expert knowledge. Furthermore, it can learn underlying information from data, which is hard to be discovered by traditional solvers. Existing data-driven CO solvers can be roughly classified into three categories based on the training procedure: (**i**) supervised learning-based solvers, (**ii**) reinforcement learning-based solvers, and (**iii**) unsupervised learning-based solvers.

Methods in the first category attempt to discover common patterns supervised by pre-computed TSP solutions. These methods typically suffer from large expenditure in computation of the pre-computed optimal or high-quality approximate TSP solutions. Apart from that, the performance of supervised learning-based methods may decrease dramatically while tackling different size TSP instances, especially for large-scale TSP. Methods in the second category rely on Markov decision process (MDP) to grow partial solutions by adding one new node whose probability is fairly high in each step or refine a feasible solution with neural network-guided local operations such as 2-opt Lin & Kernighan (1973); Andrade et al. (2012) and node swap Chen & Tian (2019); Wu et al. (2021).

However, these methods face the limitation of the ability of capture the multimodal nature of the problems Khalil et al. (2017); Gu et al. (2017) and the challenge of scalability on large TSP instances, since they suffer from the unavoidable sparse reward problem and sample efficiency issues. Methods in the third category focus on various aspects to improve their performance without relying on pre-computed data. Instead, they try to train their models with unlabeled data and compensate by specific losses or operations Karalias & Loukas (2020); Sun et al. (2022); Wang et al. (2022); Min et al. (2023); Sun et al. (2023). However, these methods also suffer from the instability and face additional complexity compared with SL and RL based methods in training since there is no label in this process.

To circumvent the above limitations, we propose a semi-supervised learning-based framework, which alleviates the strong assumption of data setting and enjoys the merits of benefiting from labeled optimal solution and utilizing abundant unlabeled dataset. We instantiate the semi-supervised framework in a generative diffusion model. The network architecture is built upon a specific transition matrix and a tailored graph neural network. Together with the dedicated design of the training procedure of diffusion model in semi-supervised manner, our method shows significant improvement compared with the state-of-the-art methods. Furthermore, extensive experiments well demonstrate the effectiveness of each component in our design. We highlight the following contributions:

- We set up a semi-supervised TSP framework to make up for the shortcomings of TSP in learning paradigm.
- We propose a diffusion model training strategy in semi-supervised manner.
- This method achieves state-of-the-art results and pushes the boundary of large-scale TSP.

## 2 RELATED WORK AND BACKGROUND

Our method is related to learning-based combinatorial optimization solver, semi-supervised learning, and the theory of diffusion model. The following paragraphs review relevant prior works.

### 2.1 LEARNING-BASED TSP SOLVER

**Supervised Learners.** Vinyals et al. (2015) first introduced a supervised learning-based algorithm, the Pointer Network, to predict a TSP solution. Nowak et al. (2017) proposed a supervised approach which can predict a heatmap of the solution space by training a graph neural network (GNN), and then utilizing beam search to convert the heatmap to a valid tour. Li et al. (2018) and Joshi et al. (2019) followed this idea, instead of continuing to train a GNN, they trained a graph convolutional network (GCN) to build a heatmap which included the possibility of each edge to be included in the optimal solution of a TSP. Recently, Fu et al. (2021) introduced a supervised learning algorithm in which a small-scale model is trained in supervised manner and then used repetitively to build heatmaps for much larger size TSP instances via a series of techniques including graph sampling, converting and heatmaps merging. Besides, Sun & Yang (2023) broadened the current scope of neural solvers for CO problems by proposing a graph-based diffusion framework, in which the training process relies on the stable supervised manner. These SL-Based methods are incredibly costly to generate ground truth tour on large scale graphs, due to the fact that they require a large number of pre-computed TSP solutions.

**Reinforcement Learners.** Bello et al. (2016) were the first to solve CO problems by combining neural networks and reinforcement learning (RL). They implemented an actor-critic RL architecture, in which the total length of a tour was considered as a reward, to improve the quality of the final solution. Since then, numerous further researches have been done, including attention models Kool et al. (2018); Deudon et al. (2018), more advantageous RL algorithms Khalil et al. (2017); Ma et al. (2019); Kwon et al. (2020); Xin et al. (2021); Choo et al. (2022), better training policies Kim et al. (2022); Bi et al. (2022), for a wider range of CO problems. Recently, Qiu et al. (2022) proposed a differentiable solver which can parameterize the underlying distribution of feasible solutions and then reduce the variance of gradients by RL-Based algorithm. Apart from these learners, another typical category of RL-Based solvers is improvement heuristics learners. Unlike construction methods mentioned before, RL-Based improvement heuristics methods train neural networks to increase the quality of the current solution until there is no computational budget. Most of these methods rely on traditional local-search algorithms such as 2-opt Croes (1958) and the large neighborhood search Shaw (1997), and have been demonstrated with outstanding results by many previous works.

However, these RL-Based models encounter the sparse reward problem when dealing with large scale graphs Joshi et al.; Kim et al. (2021); Kool et al. (2018), since the reward is only decided after decoding a complete solution. This problem can cause poor generalization performance and unstable training variance.

**Other unsupervised Learners.** Besides supervised learning-based and reinforcement learning-based algorithms mentioned before, there are some other learners introduced to solve CO problems. Karalias & Loukas (2020) inspired by Erdos' probabilistic method and utilized a neural network to parameterize a probability distribution over solution spaces. Sun et al. (2022) proposed an annealed training framework for CO problems which transformed CO problems into unbiased energy-based models and selected a special penalty to make the EBM as smooth as possible. Simultaneously, Wang et al. (2022) followed a relaxation-puls-rounding approach and adopted neural networks to parameterize the relaxed solutions instead of discrete solutions. Recently, Min et al. (2023) proposed an unsupervised learning method which trained with only a small amount of unlabeled data and compensated for it by employing a special loss function. Additionally, Sun et al. (2023) utilized modern sampling strategies to leverage landscape information to provide general-purpose solvers. However, all these unsupervised solvers face additional complexity compared with SL and RL based methods in training since there is no label in this process. Nevertheless, these methods have no advantage performance in small scale CO problems.

## 2.2 SEMI-SUPERVISED LEARNING

Semi-supervised learning Van Engelen & Hoos (2020) has proven to be a powerful paradigm for learning from datasets with rare labels or boosting the performance by leveraging unlabeled data, which has been applied broadly in many fields, including representation learning Lee et al. (2013), perceptual applications Zoph et al. (2020); Liu et al. (2020), generative models Zoph et al. (2020), and so on. In the context of TSP, unlabeled instances are generated via randomly sampling 2D positions within a unit square, *i.e.*, infinite number of unlabeled instances freely accessed. However, the label (optimal tour) is expensive especially for large-scale TSPs. Accordingly, a semi-supervised learning framework become ever more necessary, in which both the underlying characteristics of data own are explored in an unsupervised fashion and optimal tours are used to provide strong guidance via full supervision. However, to the best of our knowledge, the semi-supervised regime has not been explored in the area of TSP, where we bride this gap in this article.

## 2.3 DIFFUSION MODELS

Diffusion models Sohl-Dickstein et al. (2015); Ho et al. (2020) are latent variable generative models characterized by a forward and a reverse Markov process. The forward process $q(\boldsymbol{x}_{1:T}|\boldsymbol{x}_0) = \prod_{t=1}^{T} q(\boldsymbol{x}_t|\boldsymbol{x}_{t-1})$ gradually corrupts the data $\boldsymbol{x}_0 \sim q(\boldsymbol{x}_0)$ into a sequence of increasingly noise latent variables $\boldsymbol{x}_{1:T} = \boldsymbol{x}_1, \boldsymbol{x}_2, \dots, \boldsymbol{x}_T$. The learned reverse Markov process $p_\theta(\boldsymbol{x}_{0:T})$ gradually denoise the latent variables towards the data distribution. Altogether, denoising diffusion probabilistic models are latent variable models of the following form:

$$p_\theta(\boldsymbol{x}_0) = \int p_\theta(\boldsymbol{x}_{0:T}) \, d\boldsymbol{x}_{1:T}, \quad \text{where} \quad p_\theta(\boldsymbol{x}_{0:T}) = p(\boldsymbol{x}_T) \prod_{t=1}^{T} p_\theta(\boldsymbol{x}_{t-1}|\boldsymbol{x}_t). \tag{1}$$

In order to optimize the generative model $p_\theta(\boldsymbol{x}_0)$ to fit the data distribution $q(\boldsymbol{x}_0)$, we typically optimize a variational upper bound on the negative log-likelihood:

$$\mathbb{E}[-\log p_\theta(\boldsymbol{x}_0)] \leq \mathbb{E}_q\left[-\log p_\theta(\boldsymbol{x}_{0:T}) + \log q(\boldsymbol{x}_{1:T}|\boldsymbol{x}_0)\right],$$

$$= \mathbb{E}_q\left[\sum_{t=2}^{T} D_{KL}\left[q(\boldsymbol{x}_{t-1}|\boldsymbol{x}_t, \boldsymbol{x}_0)||p_\theta(\boldsymbol{x}_{t-1}|\boldsymbol{x}_t)\right] - \log p_\theta(\boldsymbol{x}_0|\boldsymbol{x}_1)\right] + C, \tag{2}$$

where $C = \mathbb{E}_q\left[D_{KL}\left[q(\boldsymbol{x}_T|\boldsymbol{x}_0)||p(\boldsymbol{x}_T)\right]\right]$ is a constant.

**Discrete Diffusion.** Typical diffusion models Song et al. (2020); Nichol & Dhariwal (2021) operate in the continuous domain, such as the most widely applied image domain, while the tour domain in TSP is discrete. Discrete diffusion models are recently proposed for generation of discrete image bits or texts using multinomial and categorical noises Hoogeboom et al. (2021); Austin et al. (2021). Sun & Yang (2023) achieve sound results in TSP by leveraging the diffusion model, and demonstrate the better results achieved by discrete diffusion compared with a continuous one. To this end, we build our semi-supervised diffusion solver upon discrete diffusion.

# 3 METHODOLOGY

## 3.1 PROBLEM DEFINITION

Following a conventional notation Papadimitriou & Steiglitz (1998) in combinatorial optimization (CO), we define $\mathcal{X}_s$ as the space of discrete feasible solutions for a CO instance $s$, and $c : \mathcal{X}_s \to \mathbb{R}$ as the cost function for a feasible solution $\boldsymbol{x}_s \in \mathcal{X}_s$. The objective is to find the optimal solution for a given instance $s$: $\boldsymbol{x}_s^* = \arg\min_{\boldsymbol{x}_s \in \mathcal{X}_s} c(\boldsymbol{x}_s)$.

In the context of TSP, an instance $s$ contains $N$ nodes that are connected with $E$ edges, $\mathcal{X}_s = \{0, 1\}^E$ denotes the set of all tours that visit each node exactly once and return to the starting node at the end, and $c$ calculates the cost for each tour by summing up the edge weights in the tour. A solution tour $\boldsymbol{x}_s \in \{0, 1\}^E$ for instance $s$ is exactly an indicator vector for selecting a subset from $E$ edges.

**Probabilistic Formulation for TSP.** Probabilistic TSP solvers Bello et al. (2016); Sun & Yang (2023) tackle this problem by defining a parameterized conditional distribution $p_\theta(\boldsymbol{x}_s|s)$, such that the expected cost $\sum_{\boldsymbol{x}_s \in \mathcal{X}_s} c(\boldsymbol{x}_s) \cdot p_\theta(\boldsymbol{x}_s|s)$ is minimized. Unsupervised- or reinforcement-based solvers learn the generative models by exploring the underlying principle of heuristics learners, neglecting the possibly available optimal solution; Supervised learning-based solvers pose a strict assumption that the optimal solutions $\boldsymbol{x}^*$ are always available, which is not always applicable due to the prohibitive time consumption of generating optimal solution. Instead, in this article, we bridge this gap and demonstrate a semi-supervised framework towards a more general situation, *i.e.*, we can freely produce instances as many as possible but the acquisition of optimal solution (label) $\boldsymbol{x}^*$ is costly.

Let $\mathcal{S} = S^\bullet + S^\circ$ be the TSP training instances, where $S^\bullet$ and $S^\circ$ denote labeled and unlabeled set, respectively. The learning objective $\mathcal{L}$ consists of two components: maximizing the *likelihood of optimal solutions* for labeled samples and minimized *transition matrix constraints* for all samples:

$$\mathcal{L} = \mathbb{E}_{s \in S^\bullet}[-\log p_\theta(\boldsymbol{x}_s^*|s)] + \mathbb{E}_{s \in S}[\Psi(\boldsymbol{x}_s) \cdot p_\theta(\boldsymbol{x}_s|s)], \tag{3}$$

where $\Psi(\cdot)$ denotes unsupervised losses on the predict solution $\boldsymbol{x}_s$ for all training instances. This is made viable by modeling the generative process on a proxy transition matrix Min et al. (2023) instead of the raw heatmap as the most previous methods do, which improves the performance of decoding and eases the design of our unsupervised objectives. The conditional generative distribution $p_\theta(\boldsymbol{x}_s|s)$ is parameterized by diffusion model, following Sun & Yang (2023). Our framework is notably different from them in the supervision paradigm and the form of tour for generative process. Next, we introduce our semi-supervised diffusion solver. We denote $\boldsymbol{x}_0$ as the $\boldsymbol{x}_s^*$ and $\widetilde{\boldsymbol{x}}_0$ as the predicted solution $\boldsymbol{x}_s$ following the diffusion convention, and omit the conditional notations of $s$ for brevity.

## 3.2 SEMI-SUPERVISED DIFFUSION SOLVER

**Generative Process on Transition Matrix.** Generating a *valid* tour poses challenges for learning-based methods. That is, the machine learning models, such as graph neural networks, are asked for learning from discrete tours, which are typically represented by a probabilistic *heatmap* $\boldsymbol{H} \in \mathbb{R}^{N \times N}$, where $\boldsymbol{H}_{i,j}$ denotes the probability of the edge between node $i$ and $j$ belongs to the final tour. However, the heatmap itself gives no guarantee that it corresponds to a valid tour, *i.e.*, a tour visits each node exactly once and returns to the starting node at the end, and thus it may impede the effectiveness of learning the generative process.

In order to alleviate the above concern, we propose to model the generative process upon a proxy of heatmap, named *transition matrix* that is firstly proposed in Min et al. (2023). A transition matrix $\boldsymbol{T} \in \mathbb{R}^{N \times N}$ directly defines the transition between nodes: $\boldsymbol{T}[i, j]$ indicates the probability of the $j$th in a tour sequence is the $i$th node. As a result, two adjacent columns $\boldsymbol{T}[:, j]$ and $\boldsymbol{T}[:, j + 1]$ define one transition (edge) in the solution tour. Given a transition matrix $\boldsymbol{T}$, the corresponding heatmap $\boldsymbol{H}$ can now be computed as:

$$\boldsymbol{H} = \boldsymbol{T}\boldsymbol{V}\boldsymbol{T}^\top, \text{ where } \boldsymbol{V} = \begin{bmatrix} 0 & 1 & 0 & \cdots & 0 & 0 \\ 0 & 0 & 1 & \cdots & 0 & 0 \\ \vdots & \vdots & \ddots & \ddots & \vdots & \vdots \\ 0 & 0 & 0 & \ddots & 1 & 0 \\ 0 & 0 & 0 & \cdots & 0 & 1 \\ 1 & 0 & 0 & \cdots & 0 & 0 \end{bmatrix}. \tag{4}$$

$V \in \mathbb{R}^{N \times N}$ is the Sylvester shift matrix Sylvester (1909) and can be interpreted as a cyclic permutation operator that performs a circular shift. As a side benefit, after the conversion from $T$ to $H$, the Hamiltonian Cycle constraint of a tour is naturally satisfied, which eases the heatmap decoding. We refer the readers to Min et al. (2023) for more details about the advantages of the transition matrix.

**Instantiating the Diffusion Model.** We define our TSP solver as a conditional diffusion model: the forward process gradually corrupts the data $x_0$ into noise, where the $x_0$ ***is exactly the transition matrix*** $T$ generated from the optimal tour; the denoise process takes as a set of noisy variables $x_t$ and predicts the clean data $\tilde{x}_0$, conditioned on instance $s$, *i.e.*, $N$ nodes in 2D Cartesian space.

Although the typical continuous diffusion models Ho et al. (2020); Song et al. (2020) can be directly applied to discrete data by lifting the discrete input into a continuous space, previous work Sun & Yang (2023) has proven that the continuous diffusion lags behind a discrete one. Thus, we instantiate our solver with discrete diffusion. In the discrete diffusion model, multinomial noise is used to corrupt data and the forward process add noises by:

$$q(x_t|x_{t-1}) = \text{Cat}(x_t; \mathbf{p} = x_{t-1}Q_t), \text{ where } Q_t = \begin{bmatrix} (1-\beta_t) & \beta_t \\ \beta_t & (1-\beta_t) \end{bmatrix}. \tag{5}$$

$Q_t$ is the transition probability matrix at $t$-step (It should not be confused with the $T$ that defines a tour). $xQ$ is vector-matrix product where $x$ can to be understood as one-hot vectors ($\{0,1\}^{N \times N \times 2}$), converted from the original $x \in \{0,1\}^{N \times N}$. We follow Sun & Yang (2023) and also want $\prod_{t=1}^T (1-\beta_t) \approx 0$ such that $x \sim \text{Uniform}(\cdot)$. And we obtain the following $t$-step marginal and posterior at time $t-1$:

$$q(x_t|x_0) = \text{Cat}(x_t; \mathbf{p} = x_0\overline{Q}_t), \text{ with } \overline{Q}_t = Q_1 Q_2 \ldots Q_t,$$

$$q(x_{t-1}|x_t, x_0) = \frac{q(x_t|x_{t-1}, x_0)q(x_{t-1}|x_0)}{q(x_t|x_0)} = \text{Cat}\left(x_{t-1}; \mathbf{p} = \frac{x_t Q_t^{\mathsf{T}} \odot x_0 \overline{Q}_{t-1}}{x_0 \overline{Q}_t x_t^{\mathsf{T}}}\right). \tag{6}$$

During reverse process, a neural network is trained to predict the logits of a distribution $p_\theta(\tilde{x}_0|x_t)$, which is combined with $q(x_{t-1}|x_t, x_0)$ and a summation over one-hot representations of $x_0$ to obtain the following parameterization:

$$p_\theta(x_{t-1}|x_t) = \sum_{\tilde{x}_0} q(x_{t-1}|x_t, \tilde{x}_0)p_\theta(\tilde{x}_0|x_t). \tag{7}$$

**Semi-supervised Objective.** Our semi-supervised objective consists two components as shown in Equation 3. (**i**) Maximizing the likelihood of optimal solutions is equivalent to minimize the error of predicted noise $\epsilon$ Austin et al. (2021), in which the input $x_0$ used for forward corruption is just the optimal solution of instance $s$ in $S^\bullet$. However, there is no optimal solution to feed to the forward process, for the unlabeled instances $s$ in $S^\circ$. (**ii**) We introduce a reverse-only paradigm for learning a TSP diffusion model without optimal solution. Specifically, starting from a sampled noise at timestep $T$, conditioned on the node positions of instance $s$, the predicted $\tilde{x}_0$ is estimated as:

$$p_\theta(\tilde{x}_0|x_T) \propto p_\theta(\tilde{x}_0|x_\tau)q(x_\tau|x_T, \tilde{x}_0')p_\theta(\tilde{x}_0'|x_T), \tag{8}$$

where the intermediate timestep $\tau$ is sampled uniformly. Only the gradients during the second reverse process, *i.e.*, $p_\theta(\tilde{x}_0|x_\tau)$, is enabled. We also experiment multiple intermediate steps, like the denoise process during inference, and we surprisingly find that one intermediate step is enough for the unsupervised constraint, which is verified in Section 4.3. Note that we omit the condition $s$ when we discuss the diffusion model. Subsequently, the predicted $\tilde{x}_0$ (*i.e.*, transition matrix $\hat{T}$) is constrained as following unsupervised form Min et al. (2023):

$$\Psi(\tilde{x}_0) = \lambda_1 \underbrace{\sum_{i=1}^N (\sum_{j=1}^N \tilde{T}_{i,j} - 1)^2}_{\text{Row-wise constraint}} + \lambda_2 \underbrace{\sum_i \widetilde{H}_{i,i}}_{\text{No self-loops}} + \lambda_3 \underbrace{\sum_i^N \sum_j^N D_{i,j}\widetilde{H}_{i,j}}_{\text{Minimize tour distance}}, \tag{9}$$

where $\widetilde{T}$ is just the $\tilde{x}_0$ with softmax applied in column-wise, $\widetilde{H}$ is calculated using Equation 4, $D$ is the distance matrix between nodes, and $\lambda$ balance these terms. We note the first term bring one-hot constraint to the predicted transition matrix, which plays an important role in the success of our semi-supervised framework, verified in the experiments (Section 4.3). The second term induces self-loops regularization, and the third term induces weighted edge selection constraint.

**Graph Neural Network with Transition Matrix.** We employ the graph neural network (GNN) with edge gating mechanisms Joshi et al. (2019); Sun & Yang (2023) as the backbone network for our conditional diffusion model. Sun & Yang (2023) using a heatmap-based graph neural network, where the heatmap naturally defines edges for node feature aggregation. However, aggregating node feature (message passing) based on the transition matrix $\boldsymbol{T}$ is non-trivial. To this end, we introduce an effective strategy to approach it. Given the $i$th node, we first choose a column with largest probability on the $i$th row-vector $\boldsymbol{T}_t[i,:]$, where $\boldsymbol{T}_t$ is the transition matrix at current $t$-timestep: column $j = \arg\max_j \boldsymbol{T}_t[i,j]$. The edge weight for aggregation of node $i$ is the $(j+1)$ column of *edge features* within each GNN layers. In other words, transition matrix $\boldsymbol{T}_t$ determines which column to select, edge features within each GNN layers determines the actual aggregation weights.

**Denoise Schedule for Fast Inference.** It's a common practice to speed up the inference of diffusion model by reducing the steps. We use the cosine denoising schedule, following Sun & Yang (2023). Specifically, the forward process is defined on a timestep subset $\{\boldsymbol{x}_{\tau_1}, \boldsymbol{x}_{\tau_2}, \ldots, \boldsymbol{x}_{\tau_M}\}$, where $M$ denotes the number of inference steps and $\tau_i = T - \lfloor \sin(\frac{M-i}{M} \cdot \frac{\pi}{2}) \cdot T \rfloor$. Then the denoise process during inference directly models $q(\boldsymbol{x}_{\tau_{i-1}}|\boldsymbol{x}_{\tau_i}, \boldsymbol{x}_0)$.

## 3.3 DECODING STRATEGIES

During inference, we convert the predicted transition matrix $\widetilde{\boldsymbol{T}}$ to the heatmap $\widetilde{\boldsymbol{H}}$ and try to generate the final tour solution by *heatmap decoding*. Therefore, in this work, we employ three decoding strategies following previous work Sun & Yang (2023); Qiu et al. (2022): (**i**) Greedy decoding, in which all the possible edges are sorted in decreasing order by a specific evaluation criterion, and then are inserted into the partial solution if there is no conflict. Normally, this strategy can be used with 2-opt Lin & Kernighan (1973) to achieve better results. (**ii**) Sampling, a strategy from Kool et al. (2018), in which we sample multiple solutions parallelly and select the one with the best performance. (**iii**) Monte Carlo Tree Search (MCTS) followed Fu et al. (2021), where we generate a number of actions with k-opt Croes (1958) to find much higher-quality solutions. MCTS consists of four steps: initialization, simulation, selection, and back-propagation, and these steps are iterated until no improving actions can be found in the sampling pool.

By default, we use the greedy decoding + 2-opt scheme as the default decoding scheme, following Graikos et al. (2022); Sun & Yang (2023). Also note that apart from the decoding strategies we used, there are also other kinds of decoding approaches, such as the best-first local search used in the Min et al. (2023) which can explore the search space by mining the most promising node continuously.

## 4 EXPERIMENTS

We choose various scale 2D-Euclidean TSP instances to test our model, and sample nodes from a uniform distribution over the unit square randomly to generate the instances. We consider TSP-100 and TSP-500 as the main benchmark to validate the configurations choices. Additionally, we assess our solver on larger scale TSP instances with 100, 500, 1000, and 10000 nodes to illustrate its performance as well as scalability against other start-of-the-art solvers.

## 4.1 EXPERIMENTAL SETUPS

**Datasets.** The training dataset is divided into two parts: (**i**) Labeled set $S^{\bullet}$. In order to label these instances (*i.e.*, find the optimal solution for supervised training), we use the Concorde exact solver for small-scale TSPs (TSP-50/100), and the LKH-3 heuristic solver Helsgaun (2017) for large-scale TSPs (TSP-500/1000/10000). (**ii**) Unlabeled set $S^{\circ}$. Since the TSP instances are trivially synthesized, we can freely produce unlabeled instances as many as possible. However, how to balance the size of $S^{\bullet}$ and $S^{\circ}$ is remaining to be solved, which we will fully investigate in ablation studies (Section 4.3).

By default, for all TSP scales, we use the same number of labeled instances as Sun & Yang (2023), and twice numbers for unlabeled instances, *i.e.*, the ratio between $S^{\bullet}$ and $S^{\circ}$ is $1:2$. We also demonstrate the setting where the total number of instances equals with Sun & Yang (2023), to demonstrate the superiority of the model itself. For fair evaluation, we use the same test instances as Kool et al. (2018); Joshi et al. (2019); Sun & Yang (2023) for TSP-50/100 and Fu et al. (2021) for TSP-500/1000/10000.

**Evaluation Metrics.** In order to compare performance of different models effectively, we consider three metrics which are the same as the metrics in Sun & Yang (2023): average tour length (LENGTH), average relative performance drop (DROP) and total run time (TIME). LENGTH is defined as the average length of the predicted tour for each instance in the test set; DROP is the average of the relative decrease in performance in terms of the solution length compared with a baseline solver; TIME is considered as the total clock time required to generate solutions for all test instances in seconds(s), minutes(m), or hours(h).

**Model Settings.** (**i**) GNN. A graph neural network with edge gating mechanisms is adopted as the backbone network, where message passing is achieved along the edges defined upon the transition matrix $T$, as described in Section 3.2. It has 12-layers with a width of 256. We refer reader to Joshi et al. (2019) for more GNN details. (**ii**) Diffusion. We train diffusion model with $T = 1000$ and simple linear noise schedule with $\beta_1 = 10^{-4}, \beta_T = 0.02$; We inference diffusion model with $M = 50$, *i.e.*, only steps $1, 2, 5, 8, 13, 18, \ldots, 938, 969, 1000$ are used.

**Baselines** We choose several strong baselines for comparison, including traditional TSP methods and learning-based methods: (**i**) Traditional TSP methods include Concorde Applegate (2006) and Gurobi Optimization (2018), which are exact TSP solvers, 2-opt Croes (1958) and LKH-3 Helsgaun (2017), which are heuristic solver. Farthest Insertion is also included as a simple baseline for computing the performance drop. (**ii**) Learning-based methods include AM Kool et al. (2018), GCN Joshi et al. (2019), Transformer Bresson & Laurent (2021), POMO Kwon et al. (2020), POMO+EAS Hottung et al. (2021), Att-GCN Fu et al. (2021), Sym-NCO Kim et al. (2022), DPDP Ma et al. (2021), Image Diffusion Graikos et al. (2022), MDAM Xin et al. (2021), DIMES Qiu et al. (2022), DIFUSCO Sun & Yang (2023), and UTSP Min et al. (2023).

These include classical benchmarks and the state-of-the-art proposed recently. Some of these methods are designed targeting at specific scale or not have well scalability in large-scale problem, hence we divided the comparison into small- (50/100) and large-scale (500/1000/10000) TSP and choose the competitors if they do well at current scale.

## 4.2 MAIN RESULTS

We compare our method to *all* other state-of-the-art methods as far as we know. Main results are reported on Table 1 and Table 2 across various scales.

Table 1 compares ours with other approaches on the small-scale TSP-50 and TSP-100. We adopt 50 (diffusion steps) × 1 (samples) policy and 10 (diffusion steps) × 16 (samples), denoted as Greedy and 16×Sampling. With 16×Sampling, both the DIFUSCO Sun & Yang (2023) and ours achieve almost perfect prediction, zero DROP compared with the exact solution. However, in the case of one-sampling, DIFUSCO demonstrates significant DROP, while that of ours far less than theirs. We consider that this phenomenon comes from a better modeling for the underlying data distribution, owing to our semi-supervised framework that can benefit from unlabeled data.

Table 2 compares ours with other approaches on the large-scale TSP-500, TSP-1000, and TSP-10000. Since most previous probabilistic solvers, except DIMES Qiu et al. (2022) and DIFUSCO, becomes untrainable on these scales, the results of these methods are reported with models trained on TSP-100. Following Sun & Yang (2023), we also report the results with various decoding strategies, including greedy, sampling, and MCTS, as mentioned in Section 3.3. The results show that our method remarkably outperforms state-of-the-art solvers on these large scale problems, suggesting a well scalability of our methods. Moreover, for the TSP-10000 that only approximate solutions are available, the DIFUSCO with fully supervised may learn a suboptimal distribution. In contrast, ours semi-supervised paradigm offers an opportunity for achieving better results than the LKH-3 heuristic solvers. We defer this validation in the future when the exactly results can be accessible in such scale.

## 4.3 ABLATION STUDIES

In this subsection, we study the important design choices in our framework, including (**i**) the ratio between the size of labeled subset $S^\bullet$ and unlabeled subset $S^\circ$, (**ii**) the number of intermediate steps for computing unsupervised losses, (**iii**) importance of unsupervised terms, and (**iv**) trade-off between denoising inference steps and samples. We conduct ablation experiments on TSP-100 and

Table 1: **Results on small-scale TSP.** $*$ denotes the baseline for computing the DROP . $\dagger$ denotes that we only sample once in diffusion model. Some of results are taken from Fu et al. (2021); Qiu et al. (2022); Sun & Yang (2023).

| ALGORITHM | TYPE | TSP-50 | | TSP-100 | |
|---|---|---|---|---|---|
| | | LENGTH $\downarrow$ | DROP(%) $\downarrow$ | LENGTH $\downarrow$ | DROP(%) $\downarrow$ |
| CONCORDE | EXACT | 5.69$^*$ | 0.00 | 7.76$^*$ | 0.00 |
| 2-OPT | HEURISTICS | 5.86 | 2.95 | 8.03 | 3.54 |
| AM | GREEDY | 5.80 | 1.76 | 8.12 | 4.53 |
| GCN | GREEDY | 5.87 | 3.10 | 8.41 | 8.38 |
| TRANSFORMER | GREEDY | 5.71 | 0.31 | 7.88 | 1.42 |
| POMO | GREEDY | 5.73 | 0.64 | 7.84 | 1.07 |
| SYM-NCO | GREEDY | - | - | 7.84 | 0.94 |
| DPDP | $1k$-IMPROVEMENTS | 5.70 | 0.14 | 7.89 | 1.62 |
| IMAGE DIFFUSION | GREEDY$^\dagger$ | 5.76 | 1.23 | 7.92 | 2.11 |
| DIFUSCO | GREEDY$^\dagger$ | 5.70 | 0.10 | 7.78 | 0.24 |
| OURS | GREEDY$^\dagger$ | **5.69** | **0.04** | **7.77** | **0.13** |
| AM | $1k\times$SAMPLING | 5.73 | 0.52 | 7.94 | 2.26 |
| GCN | $2k\times$SAMPLING | 5.70 | 0.01 | 7.87 | 1.39 |
| TRANSFORMER | $2k\times$SAMPLING | 5.69 | 0.00 | 7.76 | 0.39 |
| POMO | $8\times$AUGMENT | 5.69 | 0.03 | 7.77 | 0.14 |
| SYM-NCO | $100\times$SAMPLING | - | - | 7.79 | 0.39 |
| MDAM | $50\times$SAMPLING | 5.70 | 0.03 | 7.79 | 0.38 |
| DPDP | $100k$-IMPROVEMENTS | 5.70 | 0.00 | 7.77 | 0.00 |
| DIFUSCO | $16\times$SAMPLING | **5.69** | **0.00** | **7.76** | **0.00** |
| OURS | $16\times$SAMPLING | **5.69** | **0.00** | **7.76** | **0.00** |

Table 2: **Results on large-scale TSP.** $*$ denotes the baseline for computing the DROP . $\dagger$ denotes that we only sample once in diffusion model. RL, SL, SSL, AS, G, S, BS, and MCTS denotes Reinforcement Learning, Supervised Learning, Semi-Supervised Learning, Active Search, Greedy decoding, Sampling decoding, Beam-search, and Monte Carlo Tree Search, respectively. Some of results are taken from Fu et al. (2021); Qiu et al. (2022); Sun & Yang (2023).

| ALGORITHM | TYPE | TSP-500 | | | TSP-1000 | | | TSP-10000 | | |
|---|---|---|---|---|---|---|---|---|---|---|
| | | LENGTH $\downarrow$ | DROP $\downarrow$ | TIME $\downarrow$ | LENGTH $\downarrow$ | DROP $\downarrow$ | TIME $\downarrow$ | LENGTH $\downarrow$ | DROP $\downarrow$ | TIME $\downarrow$ |
| CONCORDE | EXACT | 16.55$^*$ | - | 37.66 m | 23.12$^*$ | - | 6.65 h | N/A | N/A | N/A |
| GUROBI | EXACT | 16.55 | 0.00% | 45.63 h | N/A | N/A | N/A | N/A | N/A | N/A |
| LKH-3 (DEFAULT) | HEURISTICS | 16.55 | 0.00% | 46.28 m | 23.12 | 0.00% | 2.57 h | 71.77$^*$ | - | 8.8 h |
| LKH-3 (LESS TRAILS) | HEURISTICS | 16.55 | 0.00% | 3.03 m | 23.12 | 0.00% | 7.73 h | 71.79 | - | 51.27 m |
| FARTHEST INSERTION | HEURISTICS | 18.30 | 10.57% | 0 s | 25.72 | 11.25% | 0 s | 80.59 | 12.29% | 6 s |
| AM | RL+G | 20.02 | 20.99% | 1.51 m | 31.15 | 34.75% | 3.18 m | 141.68 | 97.39% | 5.99 m |
| GCN | SL+G | 29.72 | 79.61% | 6.67 m | 48.62 | 110.29% | 28.52 m | N/A | N/A | N/A |
| POMO+EAS-EMB | RL+AS+G | 19.24 | 16.25% | 12.80 h | N/A | N/A | N/A | N/A | N/A | N/A |
| POMO+EAS-TAB | RL+AS+G | 24.54 | 48.22% | 11.61 h | 49.56 | 114.36% | 63.45 h | N/A | N/A | N/A |
| DIMES | RL+G | 18.93 | 14.38% | 0.97 m | 26.58 | 14.97% | 2.08 m | 86.44 | 20.44% | 4.65 m |
| DIMES | RL+AS+G | 17.81 | 7.61% | 2.10 h | 24.91 | 7.74% | 4.49 h | 80.45 | 12.09% | 3.07 h |
| DIFUSCO | SL+G$^\dagger$+2-OPT | 16.80 | 1.49% | **3.65 m** | 23.56 | 1.90% | **12.06 m** | 73.99 | 3.10% | **35.38 m** |
| OURS | SSL+G$^\dagger$+2-OPT | **16.72** | **1.02%** | 3.83 m | **23.46** | **1.47%** | 12.33 m | **73.48** | **2.38%** | 35.82 m |
| EAN | RL+S+2-OPT | 23.75 | 43.57% | 57.76 m | 47.73 | 106.46% | 5.39 h | N/A | N/A | N/A |
| AM | RL+S | 19.53 | 18.03% | 21.99 m | 29.90 | 29.23% | 1.64 h | 129.40 | 80.28% | 1.81 m |
| GCN | SL+BS | 30.37 | 83.55% | 38.02 m | 51.26 | 121.73% | 51.67 m | N/A | N/A | N/A |
| DIMES | RL+S | 18.84 | 13.84% | 1.06 m | 26.36 | 14.01% | 2.38 m | 85.75 | 19.48% | 4.80 m |
| DIMES | RL+AS+S | 17.80 | 7.55% | 2.11 h | 24.89 | 7.70% | 4.53 h | 80.42 | 12.05% | 3.12 h |
| DIFUSCO | SL+S+2-OPT | 16.65 | 0.57% | **11.46 m** | 23.45 | 1.43% | **48.09 m** | 73.89 | 2.95% | **6.72 h** |
| OURS | SSL+S+2-OPT | **16.59** | **0.24%** | 11.63 m | **23.37** | **1.08%** | 49.20 m | **73.42** | **2.30%** | 6.73 h |
| ATT-GCN | SL+MCTS | 16.97 | 2.54% | 2.20 m | 23.86 | 3.22% | 4.10 m | 73.93 | 4.39% | 21.49 m |
| DIMES | RL+MCTS | 16.87 | 1.93% | 2.92 m | 23.73 | 2.64% | 6.87 m | 74.63 | 3.98% | 29.83 m |
| DIMES | RL+AS+MCTS | 16.84 | 1.76% | 2.15 h | 23.69 | 2.46% | 4.62 h | 74.06 | 3.19% | 3.57 h |
| DIFUSCO | SL+MCTS | 16.63 | 0.46% | **10.13 h** | 23.39 | 1.17% | **24.47 m** | 73.62 | 2.58% | **47.36 m** |
| OURS | SSL+MCTS | **16.57** | **0.12%** | 10.44 m | **23.28** | **0.69%** | 25.18 m | **73.33** | **2.17%** | 48.66 m |

TSP-500, with greedy decoding and sampling+2-OPT decoding respectively. We report the average tour length, illustrated in Table 3, 4, 5, and Figure 1.

**i. Ratio Between $S^\bullet$ and $S^\circ$.** Table 3 shows the results on various ratio between $S^\bullet$ and $S^\circ$. As the size of unlabeled set increases, the performance is getting better obviously. Due to the limit computation resource, we validate the ratio up to $1 : 10$. For fair comparison with previous supervised methods, we only use $1 : 2$ in previous main results section. We also compare our method to DIFUSCO with the same total numbers of instances, where we only use half numbers of labels, shown in the row start with $0.5, 0.5$. We see that our method performs even better at large-scale TSP-500 problem.

Table 3: The impact of the ratio between $S^{\bullet}$ and $S^{\circ}$.

| Ratios | TSP-100 | TSP-500 |
|---|---|---|
| 1 : 0 | 5.70 | 16.63 |
| 1 : 0.5 | 5.70 | 16.61 |
| 1 : 1 | 5.69 | 16.61 |
| 1 : 2 | 5.69 | 16.59 |
| 1 : 5 | 5.69 | 16.57 |
| 1 : 10 | 5.69 | 16.55 |
| 0.5, 0.5 | 5.72 | 16.64 |
| DIFUSCO | 5.70 | 16.65 |

Table 4: Ablation studies on the diffusion intermediate steps when computing unsupervised losses.

| Steps | TSP-100 | TSP-500 |
|---|---|---|
| N/A | 5.70 | 16.63 |
| 1 | 5.69 | 16.59 |
| 2 | 5.69 | 16.58 |
| 4 | 5.69 | 16.59 |
| 8 | 5.69 | 16.57 |

Table 5: Ablation studies on the importance of different unsupervision loss terms defined in Equation 9.

| Row | Loop | Dist | TSP-100 | TSP-500 |
|---|---|---|---|---|
| ✓ | | | 5.69 | 16.60 |
| ✓ | ✓ | | 5.69 | 16.60 |
| | | ✓ | 5.70 | 16.63 |
| | ✓ | ✓ | 5.71 | 16.62 |
| ✓ | ✓ | ✓ | 5.69 | 16.59 |

**ii. Intermediate Steps.** Table 4 shows the impact of various intermediate steps when computing unsupervision losses in Equation 9. N/A indicates no unsupervision losses are applied. For both other experiments, the gradient is enabled only when the last denoising process. We surprisingly find the unsupervision paradigm with only one intermediate step achieves sound results, for which we set step $= 1$ in our default configuration.

**iii. Importance of Unsupervised Terms.** Table 5 demonstrates the ablation studies on the importance of different unsupervised terms in Equation 9. We note that the performance on the small-scale TSP-100 is almost saturated, the influences on average length may not obvious. However, the improvement brought by these loss terms is remarkable on TSP-500.

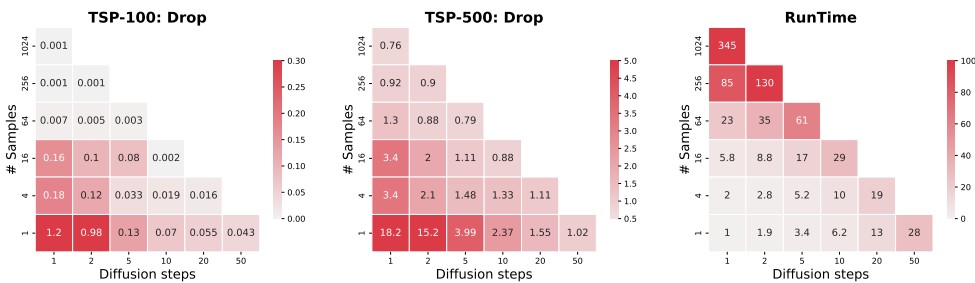

Figure 1: Ablation studies on the tradeoff between diffusion inference steps and samples. **Left** and **Middle**: The performance drop on the TSP-100 and TSP-500. **Right**: The runtime ratio which we calculate on the average of all instances in TSP-100 and TSP-500.

**iv. Diffusion Steps and Samples.** It's good practice to accelerate diffusion inference by reducing the denoising steps. And there is also an effective way to get better results by sampling multiple times and choose the best one. To this end, for the time sensitive TSP, it's importance to balance the diffusion steps and samples. We dive into this tradeoff and report the results in Figure 1. The diffusion steps we interested are $1, 2, 5, 10, 20, 50$.

## 5 CONCLUSIONS

In this article, we propose a novel semi-supervised diffusion solver for TSP towards a more general situations for large scale problem, where we can freely access to plenty of instances but few of them are labeled with optimal solutions. This semi-supervised paradigm is more flexible to the dataset settings, while the previous fully supervised or unsupervised methods pose strong assumptions to the settings. Built upon a special transition matrix and a tailored graph neural network, our method significantly improves the performance compared with the most recently state-of-the-art methods. Extensive ablation studies well demonstrate the reasonability of each component in our design. In the future, we will go further and extend the semi-supervised paradigm to other combinatorial optimization problems.

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
