# OpenReview forum: "Semi-supervised Diffusion Solver for Travelling Salesman Problem"
_ICLR.cc/2024/Conference — ICLR 2024 Conference Withdrawn Submission_

### Official Review · Reviewer_FvPj · 2023-10-18

**Soundness:** 2 fair
**Presentation:** 3 good
**Contribution:** 2 fair
**Rating:** 3
**Confidence:** 3

**Summary:**

This paper proposes a semi-supervised diffusion solver for TSP. Technically, it combines DIFUSCO [1] (i.e., supervised discrete diffusion model) and UTSP [2] (i.e., unsupervised loss) to achieve a semi-supervised objective function. To ease the decoding, a proxy of heatmap - transition matrix $T$ is used as the initial state for the forward pass. Experimental results on large-scale TSP demonstrate its superiority over several baselines.

**Strengths:**

* This paper is well-written. The empirical results look good.
* The studied semi-supervised setting is new and practical for COPs.
* The idea of modeling the generative process upon a transition matrix is interesting.

**Weaknesses:**

* The scope of this paper is limited to TSP, while DIFUSCO [1] could solve MIS and TSP. It is suggested to include another problem, such as MIS or CVRP.
* The technical contribution seems to be limited since this work mostly builds upon DIFUSCO [1] and UTSP [2].
* What are the pros of your method and the proposed semi-supervised setting compared with UTSP [2] if only TSP is solved? The training of UTSP seems to be very efficient, i.e., ~2 hours on TSP1000 without any labels. Could you report your training overhead? Moreover, UTSP should be another baseline, while I did not find it in Tables 1 and 2.
* Recent studies [3] find data augmentations could greatly ease the labeled burden of supervised learning (SL) methods in neural combinatorial optimization. Is it possible that (SL) DIFUSCO with data augmentations is comparable to (Semi-SL) your method, or even more powerful?
* Lack of analyses, any visualization (e.g., of the denoising process or obtained solution) would be better.
* The results on TSPLIB should be reported.
* Minor:
  * Better to use some figures for illustration (e.g., overview).
  * The citation format is not correct (e.g., `~\citep` and `~\citet`).
  * The reproducibility is not clear. No source code is provided.

[1] DIFUSCO: Graph-based Diffusion Solvers for Combinatorial Optimization. In NeurIPS 2023.
[2] Unsupervised Learning for Solving the Travelling Salesman Problem. In NeurIPS 2023.
[3] Data-efficient Supervised Learning is Powerful for Neural Combinatorial Optimization. Submitted to ICLR 2023.

----

**Overall,** I don't see the benefits of combining supervised DIFUSCO with unsupervised UTSP. To be more clear, (1) compared with DIFUSCO, the proposed method cannot solve other COPs; (2) compared with UTSP, your training overhead seems to be much higher. Although with slight performance improvement, I think the current contribution is not enough for ICLR. Providing further analyses, highlighting the advantage of your method and setting, and adding another COP may significantly enhance the paper's strength. I vote for rejection.

**Questions:**

* Due to the usage of $T$, could the denoising of your method directly output a feasible solution (without search)? If this is the case, could this strength help the denoising process, such as reducing the total inference time?
* What is the total run time (i.e., TIME) on TSP50 and TSP100?
* For Table 3, why does your method still outperform DIFUSCO at the ratio of 1:0?
* Since generating unlabeled TSP instances is extremely easy, it is interesting to see your performance when the ratio scales up to 1:100 or even larger.
* In each batch of the training, assume you have some labeled and unlabeled instances. So for the unlabeled instances, only the second part of Eq. (3) is activated? Could you add a pseudo-code?

---

### Official Review · Reviewer_JPUZ · 2023-10-25

**Soundness:** 2 fair
**Presentation:** 3 good
**Contribution:** 2 fair
**Rating:** 5
**Confidence:** 5

**Summary:**

This paper proposed a semi-supervised learning method for TSP.

They use discrete diffusion to build the transition matrix $\mathbb{T}$ and build the heatmap using the $\mathbb{T} \rightarrow \mathcal{H}$ proposed in Min et al.

Compared with DIFUSCO (which is a SL model),  they show they achieve better performance by adding the UL loss. The ablation study further show that the UL loss can significantly improve the performance.

Reference:

Min, Yimeng, Yiwei Bai, and Carla P. Gomes. "Unsupervised Learning for Solving the Travelling Salesman Problem." arXiv preprint arXiv:2303.10538 (2023).

Sun, Zhiqing, and Yiming Yang. "Difusco: Graph-based diffusion solvers for combinatorial optimization." arXiv preprint arXiv:2302.08224 (2023).

**Strengths:**

TSP holds a pivotal position in the realms of computer science, operations research, and combinatorial optimization. Investigating methodologies for constructing data-driven solvers for TSP is important for the Machine Learning for Combinatorial Optimization (ML4CO) community.

This paper studies SSL for TSP,  it combine the diffusion model in Graikos et al. [1] and unsupervised TSP in Min et al. [2].


Numerous articles have been published on the topic of supervised learning (SL)-based TSP solvers. Semi-supervised learning (SSL) can be viewed as an augmentation of these SL-based methodologies. Contrarily, unsupervised learning (UL)-based TSP solvers represent a new area of research.

This manuscript integrates both SL and UL approaches, demonstrating that the introduction of UL loss significantly enhances performance. Given that the UL loss concept was recently introduced by [2], this represents the (first?) application of SSL techniques to the TSP domain.


Reference

[1]: Graikos, Alexandros, et al. "Diffusion models as plug-and-play priors." Advances in Neural Information Processing Systems 35 (2022): 14715-14728.

[2]: Min, Yimeng, Yiwei Bai, and Carla P. Gomes. "Unsupervised Learning for Solving the Travelling Salesman Problem." arXiv preprint arXiv:2303.10538 (2023).

**Weaknesses:**

1. The novelty is mediocre. It's a combination of graph diffusion and UTSP.
2. only limited to TSP, but this is not a big issue, as TSP is one of the most studied problems in optimization and has various real-world applications.

**Questions:**

I am open to raise my rating if some of the following concerns are addressed:

1. In Table 3,4,5, TSP-100 results (the distance) does not match with Table 1, 5.69 should correspond to TSP-50.

2. In table 3, the authors achieve 16.55 on TSP-500 using ratio 1:10, however, in Table 2, it reports 16.72, why different?

Minor:

1. Table 2, DIFUSCO's time for TSP 500 should be 10.13m instead of 10.13h?
2. In abstract, the author use Travelling Salesman at first and Traveling Salesman and end, Travelling/Traveling should be consistent.

---

### Official Review · Reviewer_9n2o · 2023-10-29

**Soundness:** 2 fair
**Presentation:** 2 fair
**Contribution:** 2 fair
**Rating:** 3
**Confidence:** 5

**Summary:**

This paper proposes a novel semi-supervised diffusion solver for the TSP problem. The semi-supervised training method tackles the limitations of previous methods, including reinforcement learning, supervised learning and unsupervised learning methods. It achieves a clear improvement on large-scale TSP problems against the SOTA neural method.

**Strengths:**

1. The problem addressed by this paper is significant, combining both the labeled and unlabeled data is promising in creating a commercial-level neural solver.
2. The way to train the diffusion model in an unsupervised way is inspiring, which looks to me a promising direction in the future.
3. The proposed method can show a consistent improvement under different search algorithms.

**Weaknesses:**

1. Though the problem studied by this paper is significant, I'm afraid the proposed method makes a quite incremental technical contribution. Equation 8 is the only novelty in this paper and the overall framework simply combines the method of DIFUSCO and UTSP without any logic behind it. It is probably fine to call the proposed method a semi-supervised learning method, but simply combining a supervised learning objective and an unsupervised learning objective does not reveal any nature of the problem. I hope the authors to answer why they choose DIFUSCO + UTSP rather than other combinations (for example, Att-GCN + UTSP, DIFUSCO + Erdos goes neural) from the perspective of CO or TSP.
2. The experimental results are not convincing, with actually very small improvements. In both Table 1 and Table 2, it is hard for me to say the proposed method shows a significant advantage against DIFUSCO, considering the more training samples used, which I believe will unavoidably lead to a longer training time.
3. The proposed method still uses lots of labeled samples in the experiment, and this is in conflict with the claimed limitation of the supervised learning methods `These methods typically suffer from large expenditure in computation of the pre-computed optimal or high-quality approximate TSP solutions`. Now the question is, does your method really resolve this limitation of supervised learning methods? I think authors should set a goal to use less than 10% labeled instances to achieve a comparable performance with DIFUSCO trained with 100% labeled instances and much better performance than DIFUSCO trained with 10% labeled instances as well.
4. The presentation is also bad, there are many grammar errors and typos, for example, the time for DIFUSCO on TSP-500 in Table 2 should be in minutes rather than in hours. Also, I cannot find the results of DIMES and UTSP in Table 1, which is confusing.

**Questions:**

See the questions raised in the weakness session. Besides, I think [1] is just a sampling-based method without any training, so it should not be categorized as an unsupervised method. And it definitely shall not share the same limitations with other unsupervised learning method.

[1] Haoran Sun, Katayoon Goshvadi, Azade Nova, Dale Schuurmans, and Hanjun Dai. Revisiting sampling for combinatorial optimization. 2023.

---

### Official Review · Reviewer_r8xE · 2023-11-05

**Soundness:** 4 excellent
**Presentation:** 2 fair
**Contribution:** 2 fair
**Rating:** 6
**Confidence:** 3

**Summary:**

Authors propose semi-supervised diffusion solver for traveling salesman problem (TSP). Mainly, there are three technical contributions. First, authors adopt discrete diffusion model from DIFUSCO (Sun and Yang). The choice of diffusion process, and decoding strategies are mostly followed from DIFUSCO. Second, authors Adopt transition matrix-based heatmap formulation from UTSP (Min et al). This heatmap-based formulation enabled UTSP to be very sample-efficient and work with unsupervised learning. This sample-efficiency advantage seems to transfer to semi-supervised setting. Third, authors apply the combination of these techniques into semi-supervised setting. For unlabeled instances, authors propose a formulation where two reverse processes are used. On stand benchmark configurations, the proposed method demonstrates consistent improvement over baselines.

**Strengths:**

Quality: The proposed method combines most recent techniques for Neural TSP solvers. Experimental results show quite consistent improvement over baseline methods; gap reduction is often reduced to 1/2~1/3 from a strong baseline method DIFUSCO. Ablation studies provide good justification for both of technical innovations: semi-supervised learning, and heatmap-based formulation on top of DIFUSCO.

Originality & Significance: While authors build upon DIFUSCO and UTSP, it has been unclear whether the combination of these techniques, combined with semi-supervised learning, actually works. Hence, the paper provides readers a convincing recipe for building state-of-the-art neural TSP solver. This will help the research community to establish a best practice. It is also good to confirm that Discrete Diffusion model from DIFUSCO continues to work well on another heatmap-based formulation. This helps the community to be more confident about discrete diffusion-based methods, and motivate people to follow-up.

Clarity: Overall technical strategy of the paper is easy to understand.

**Weaknesses:**

While one of the key contributions of this paper is the use of semi-supervised learning, I find its formulation is not sufficiently explained. The key equation (8) for semi-supervised loss is presented without much motivation or justification. It seems like there are two diffusion processes, one for prediction and the other for training, but I wasn't sure.

Much of the training & inference recipe comes from Sun & Yang. In order to highlight authors' contribution, I would suggest to focus on one setting which worked best (or reasonably) in Sun & Yang (say, MCTS), and spend more pages on deeper analysis of semi-supervised learning, which is the main contribution of the paper. Table 3 does show that semi-supervised learning improves upon supervised baseline,  but more could've been explore to strengthen the unique contribution of the paper. For example, how does semi-supervised learning help in generalization to larger instances? For ex, can we use supervised labels of TSP-50 to generalize to TSP-1000?

**Questions:**

For semi-supervised instances, is $x_T$ sampled from a uniform distribution? Is the model trained to predict its own output?

In Section 3.2, "T[i,j] indicates the probability of the j-th in a tour sequence is the ith node" seems to be a grammatical error? Overall, I hope authors explained the heatmap and transition matrix better, at least in the appendix, because it is an important component of this work.

---

### Comment · Area_Chair_zFvv · 2023-12-02
**Regarding Submission Response**

It has come to our notice that the authors of the submission in question have not submitted their response within the stipulated timeframe. This lack of response is significant as it potentially indicates an inability or unwillingness to engage with the review process.

Given this situation, and in line with the standards of the review process, it seems appropriate to consider this submission as a potential rejection.

AC